

# Spectral Induced Polarization imaging to investigate an ice-rich mountain permafrost site in Switzerland

Theresa Maierhofer[1,2], Christian Hauck[2], Christin Hilbich[2], Andreas Kemna[3], Adrián Flores-Orozco[1]

[1]Department of Geodesy and Geoinformation, TU-Wien, Vienna, 1040, Austria
[2]Department of Geosciences, University of Fribourg, Fribourg, 1700, Switzerland
[3]Department of Geosciences, University of Bonn, Bonn, 53121, Germany

*Correspondence to*: Theresa Maierhofer (theresa.maierhofer@tuwien.ac.at)

**Abstract.** Spectral induced polarization (SIP) measurements were collected at the Lapires talus slope, a long-term permafrost monitoring site located in the Western Swiss Alps, to assess the potential of the frequency dependence (within the frequency
range of 0.1-225 Hz) of the electrical polarization response of frozen rocks for an improved permafrost characterization. The aim of our investigation was to (a) find a field protocol that provides SIP imaging data sets less affected by electromagnetic coupling and easy to deploy in rough terrains, (b) cover the spatial extent of the local permafrost distribution, and (c) evaluate the potential of the spectral data to discriminate between different substrates and spatial variations in the volumetric ice content within the talus slope. To qualitatively assess data uncertainty, we analyze the misfit between normal and reciprocal (N&R)
measurements collected for all profiles and frequencies. A comparison between different cable setups reveals the lowest N&R misfits for coaxial cables and the possibility to collect high-quality SIP data in the range between 0.1-75 Hz. We observe an overall smaller spatial extent of the ice-rich permafrost body compared to its assumed distribution from previous studies. Our results further suggest that SIP data help to improve the discrimination between ice-rich permafrost and unfrozen bedrock in ambiguous cases based on their characteristic spectral behavior, with ice-rich areas showing a stronger polarization towards
higher frequencies in agreement with the well-known spectral response of ice.

## 1 Introduction

Mountain permafrost regions are highly sensitive to climate changes, with significant implications for the hydrological regimes. Water reservoirs and sources in many mountain ranges in the world are threatened by changes of the cryosphere; and thus, knowledge of the ground ice and water content in permafrost regions is critical for the estimation of water storage
capacities and future water supplies (e.g., Arenson and Jakob, 2010; Halla et al., 2021; Harrington et al., 2018; Langston et al., 2011; Rangecroft et al., 2016; Schrott, 1998). Monitoring of the thermal state of permafrost has become an essential task in mountain regions to better understand the dynamics of mountain permafrost, for instance through borehole ground temperature networks on a global (Global Terrestrial Network of Permafrost, GTNP; Biskaborn, 2016) and regional (Swiss Permafrost Monitoring Network (PERMOS, 2021)) scale. However, from borehole temperature monitoring alone we cannot differentiate
between unfrozen and frozen water content, as we can observe water in the liquid phase even at negative temperatures (e.g.,



Harris et al., 1988). Additionally, the ground ice content is difficult to assess quantitatively and depends on many parameters such as the permafrost landform and substrate as well as past climate evolution and local topoclimatic effects (e.g., Kenner et al., 2019). Ex-situ analysis of ice-cores recovered from drilling of boreholes (e.g., Haeberli et al., 2018; Krainer et al., 2015; Wang et al., 2018) and nuclear well-logging can help to quantify porosity and ice content at specific borehole locations (e.g.,

Scapozza et al., 2015a; Vonder Mühll & Holub 1992). However, boreholes are costly and difficult to install at high-mountain permafrost sites and the information obtained is limited to discrete locations.

Geophysical methods have emerged as important techniques in alpine permafrost investigations, as they provide quasi-continuous information about subsurface properties and heterogeneities (e.g., Hauck et al., 2011; Hilbich et al., 2008; Mollaret et al., 2019; Steiner et al., 2021; Wagner et al., 2019). Hence, geophysical information supports and interconnects spatially

sparse borehole data and can provide essential information about the temporal evolution of subsurface permafrost characteristics (e.g., Hauck, 2013; Scott et al., 1990). In particular, electrical resistivity tomography (ERT) has become a routine tool for the monitoring of active layer dynamics and the internal structure of permafrost landforms, taking into account the controls of temperature on electrical resistivity (e.g., Coperey et al., 2019a; Hauck, 2002; Hauck et al., 2011; Hilbich et al., 2008, 2009; Keuschnig et al., 2017; Krautblatter et al., 2010; Mollaret et al., 2019; Parkhomenko, 1982; Supper et al., 2014).

The discrimination between unfrozen and frozen water content could be improved with the help of ERT, especially in monitoring applications although uncertainties remain (e.g., Hauck, 2002; Hauck and Kneisel, 2008; Oldenborger and LeBlanc, 2018). In case of unfrozen materials low resistivity values are observed due to electrolytic conduction taking place along the water-interconnected pores and along the electrical double layers (EDLs) formed at the interface between water and mineral surfaces (e.g., Leroy et al., 2008; Revil and Glover, 1998; Ward, 1990; Waxman et al., 1968). Upon freezing, the

mobility of the ions is reduced and conductive pathways break down increasing the electrical resistivity for frozen soils and rocks (e.g., Hauck, 2002).

In many practical applications, the interpretation of the subsurface electrical resistivity is difficult, since air, rock matrix and ice can result in similar resistivity values. Refraction Seismic Tomography (RST) is commonly used as an additional method to improve the quantification of the ice content within the subsurface, as the acoustic velocity of air and ice differs by an order

of magnitude (Hausmann et al., 2007; Hilbich, 2010). Within the so-called 4-phase model (4PM, Hauck et al., 2011), electrical resistivity and P-wave velocity data sets are combined to estimate ice, water and air contents from petrophysical relations. Recently, Wagner et al. (2019) implemented a joint inversion algorithm that iteratively solves for the petrophysical parameters from ERT and RST datasets based on the 4PM. This algorithm has demonstrated an improved ability to estimate relevant parameters (such as water and ice content) for a variety of permafrost sites (Mollaret et al., 2019) and for time-lapse monitoring

applications (Steiner et al., 2021).  However, the interpretation of geophysical signatures of air, ice and rock matrix is still open to discussion.

The induced polarization (IP) method has emerged as a promising method in hydrogeological studies due to its sensitivity to pore-space geometrical characteristics (e.g., Binley et al., 2015; Binley and Slater, 2020; Kemna et al., 2012; Revil et al., 2012; Weller et al., 2013). While the resistivity method measures the medium's ability to conduct electric current (transport of



charges) in terms of transfer resistances, the (frequency-domain) IP method also measures its ability to electrically polarize (local separation of charges and thus electrical energy storage) in terms of electrical impedances (e.g., Binley et al., 2005; Kemna et al., 2012; Revil and Skold, 2011; Sumner, 1976). The build-up of charge concentration gradients responsible for the polarization effects measured in IP takes place in the vicinity of EDLs at grain-fluid interfaces as well as pore constrictions (e.g., Bücker et al., 2019; Leroy et al., 2008; Revil, 2012; Revil and Florsch, 2010; Schwarz, 1962). IP measurements can be

performed at different frequencies in the mHz-kHz range in the so-called spectral IP (SIP) method to quantify and model the frequency dependence of the subsurface electrical properties and link them to lithological, textural, hydraulic, or geochemical soil/rock properties (see Binley et al., 2015; Kemna et al., 2012; Revil et al., 2012b; Weigand et al., 2017 for further details). Laboratory studies have demonstrated the sensitivity of SIP measurements to changes in temperature above the freezing point (e.g., Binley et al., 2010; Zisser et al., 2010) and in frozen samples (e.g., Coperey et al., 2019; Kemna et al., 2014a; Revil et

al., 2019).

Laboratory studies have addressed the polarization response of soils and rocks under freezing conditions (e.g., Kemna et al., 2014a; Olhoeft, 1977; Stillman et al., 2010; Wu et al., 2017). In the context of permafrost, Duvillard et al. (2018) and Coperey et al. (2019) investigated in the laboratory the effect of temperature (between -15 °C and +20 °C) on the SIP response of various types of porous media (e.g., sandstone, granite, soil and sands) for frequencies between 0.01 Hz and 40 kHz. Revil et

al. (2019) extended the investigations to porous media with metallic particles and Abdulsamad et al. (2019) to account for conductive minerals related to graphitic schists. Both works used an extended formulation of the Stern-layer polarization mechanism (e.g., Leroy et al., 2008; Revil and Florsch, 2010) based on an exponential freezing curve (Duvillard et al., 2018) to infer the distribution of temperature in permafrost rocks from complex conductivity data (e.g., Duvillard et al., 2021). Kemna et al. (2014a) presented the SIP freezing/thawing behaviour observed on sandstone and limestone samples with the

interpretation of seeing the breakdown of membrane polarization at frequencies below 100 Hz upon ice crystallization reflecting the Gibbs-Thomson dependence upon temperature. Additionally, Kemna et al. (2014a) and Coperey et al. (2019) report a high-frequency polarization response in measurements above 10 Hz, which they relate to the dielectric relaxation of ice associated with Bjerrum defects, as also suggested by Stillman et al. (2010). The polarization of pure ice (Auty and Cole, 1952) and ice-silicate mixtures (Stillman et al., 2010) has been characterized in the laboratory; while Grimm and Stillman

(2015) report measurements at the field scale under well-controlled conditions in a permafrost tunnel in frozen silts. Using laboratory observations, Grimm and Stillman (2015) derived ice content estimates from the resistivity frequency effect, a measure of the IP effect, and demonstrated the applicability of the SIP method in permafrost investigations. However, only very few studies investigated the applicability of the IP method in mountainous regions (e.g., Duvillard et al., 2021, 2018; Mudler et al., 2019).


The main challenge regarding the conduction of field SIP measurements corresponds to the contamination of the data at frequencies above 1 Hz by electromagnetic (EM) coupling (e.g., Flores Orozco et al., 2021; Pelton et al., 1978; Zimmermann et al., 2019). EM coupling may result in self-induction of the near subsurface materials, cross-talking between cables, and the



capacitive coupling in the contacts between cable-electrode and electrode-ground (e.g., Flores Orozco et al., 2021; Zhao et al.,
2013; Zimmermann et al., 2008, 2019). Hence, IP field surveys at higher frequencies are commonly based on the use of
separated cables connecting voltage and current electrodes (e.g., Dahlin et al., 2002) to minimize cross talking, leading to
longer field procedures than for single cable layouts. The challenges in the data acquisition have limited the application of the
SIP method in periglacial environments (Bazin et al., 2019; Doetsch et al., 2015; Duvillard et al., 2021, 2018; Grimm and
Stillman, 2015) and, to our knowledge, no study has investigated to date the influence of EM coupling and the reliability of
frequency-domain SIP imaging data sets in high-mountain permafrost environments. Nonetheless, such investigations are
critical to evaluate the applicability of the SIP method on permafrost terrain and to extend the capabilities of current permafrost
monitoring systems based on the resistivity method.

In this study, we aim at investigating the frequency dependence in SIP imaging results for measurements conducted at the
Lapires field site, a high-alpine talus slope in the Swiss Alps. The talus slope represents a typical mountain permafrost landform
with a coarse-blocky surface and high, heterogeneously distributed ground ice contents. The selected study area is a well-
characterized site with extensive information available for the interpretation of the SIP results (e.g., Delaloye, 2004; Hilbich,
2010; Mollaret et al., 2019; Scapozza et al., 2015a). Particular emphasis in our study is on the analysis of the quality of the
SIP data and an efficient field procedure that provides robust and high-quality SIP imaging data sets less affected by EM
coupling while being easy to deploy in rough terrains. To achieve this, we compare data collected with different cables
(multicore, coaxial) and different cable layouts. We propose procedures for the collection and processing of SIP data, and
discuss the spatial extent of the ice-rich permafrost at the site as resolved by means of ERT and SIP imaging. Additionally, we
investigate the frequency dependence of the observed field SIP signatures and compare it with SIP laboratory measurements
on rock samples from the site and data available in the literature. We hypothesize that the use of coaxial cables improves the
SIP data quality and that the SIP method helps to differentiate between different subsurface substrates and thermal conditions
at the investigated site.

## 2 Spectral induced polarization and complex resistivity

The frequency-domain IP method, also known as complex resistivity (CR) or complex conductivity (CC) method, uses four-
electrode arrays to measure the electrical impedance ($|Z|$). A sinusoidal current is injected into the ground through two
electrodes, while a second electrode pair is used to measure the resultant voltage, with the impedance given by the amplitude
ratio and phase-shift between the measured voltage and the injected current. Current injections can be performed at different
frequencies, commonly in the mHz-kHz range, to quantify the frequency dependence of the measurements (i.e., impedance
spectra), referred to as spectral IP (SIP) method. Imaging applications use tens to hundreds of electrodes to collect thousands
of electrical impedance readings, which, by the use of inversion algorithms (e.g., Binley and Kemna, 2005), permit to resolve
for variations of the complex electrical resistivity ($\rho^*$) (or its inverse, the complex electrical conductivity ($\sigma^*$)) in the



subsurface. The complex resistivity can be expressed in terms of the real ($\rho'(\omega)$) ($\Omega$m) and (negative) imaginary ($\rho''(\omega) > 0$) ($\Omega$m) components, such as (e.g., Binley and Slater, 2020; Wait, 1984):

$$\frac{1}{\sigma^*(\omega)} = \rho^*(\omega) = \rho'(\omega) - i\rho''(\omega) = |\rho^*(\omega)|e^{i\phi(\omega)}, \tag{1}$$

where $\omega$ is the angular frequency ($\omega = 2\pi f$, with $f$ being the excitation frequency) and $i$ represents the imaginary unit ($i^2 = -1$). The real component represents energy loss associated with conduction (dc resistivity), while the imaginary component

represents energy storage associated with polarization (capacitive property). The complex resistivity can also be expressed in terms of its magnitude ($|\rho(\omega)|$) ($\Omega$m) and phase shift ($\phi$) (rad):

$$|\rho^*(\omega)| = \sqrt{\rho'(\omega)^2 + \rho''(\omega)^2}, \tag{2}$$

$$\phi(\omega) = \arctan(\frac{-\rho''(\omega)}{\rho'(\omega)}). \tag{3}$$

We note that due to the low phase values typical in geophysical applications, we can assume that $\rho' \approx |\rho^*|$. Accordingly, the

in-phase resistivity ($\rho'$) is practically equal to the resistivity magnitude, which is resolved by ERT.

There is no fundamental reason and no loss of generality in choosing $\rho^*$ or $\sigma^*$ for the description (e.g., Wait, 1984). We opted in our study to present imaging results in terms of $\rho^*$ to better compare it to previous ERT investigations at the site, which are expressed in terms of the electrical resistivity. To compare the frequency dependence of the IP response (i.e., the SIP signatures) observed in our study with the laboratory results under freezing conditions presented in Coperey et al. (2019) and

Revil et al. (2019), we inspect the SIP signatures in terms of $\sigma'$ and $\sigma''$.

**2.1 Polarization processes in porous media and in the presence of ice**

In the presence of (liquid) pore water in a metal-free porous rock, polarization effects for frequencies below 1 kHz are generally related to ionic polarization in the EDL at the pore water – solid matrix interface (e.g., Kemna et al., 2012). Polarization mechanisms have been expressed in terms of the polarization of the Stern layer, i.e., the inner part of the EDL, (e.g., Leroy et

al., 2008; Revil, 2012), the diffuse layer, i.e., the outer part of the EDL (e.g., Dukhin et al., 1974), and the membrane polarization (e.g., Hördt et al., 2016; Marshall and Madden, 1959; Vinegar and Waxman, 1984). The study of Bücker et al. (2019) demonstrates that such models of diffuse layer and membrane polarization are equivalent and quantitatively describe the same response.

Below the freezing point, during the phase change from liquid water (further referred to as simply "water") to ice, the salinity of the liquid pore water phase is expected to increase as saline water begins to crystallize only below the eutectic temperature (∼ -21°C) due to the low solubility limit of salt in ice (e.g., Hobbs, 1974; McKenzie et al., 2007). Thus, above the eutectic temperature and below the freezing temperature, the salt remains in the pore water (e.g., Coperey et al., 2019; Revil et al., 2019) increasing the fluid conductivity. Once the ice starts to form, the change in water content with temperature is gradual

and follows a freezing curve (or thawing curve, where no super-cooling is present) (e.g., McKenzie et al., 2007). The unfrozen



water includes residual water films (bound water) at the ice and mineral surfaces, as well as water in smaller pores with sufficiently reduced melting point according to the Gibb-Thomson effect (see, e.g., Dash et al., 2006). Kemna et al. (2014a) attributed the SIP freezing/thawing behavior in saturated rocks to both liquid water films and the presence of unfrozen and frozen pores as governed by the Gibbs-Thomson effect, leading to the interpretation of the observed responses as a breakdown
of electrochemical (membrane) polarization with ongoing freezing from larger to smaller pores (lower-frequency response) along with the continuous increase of ice polarization (higher-frequency response). Coperey et al. (2019) extended the Stern-layer model developed by Revil (2012, 2013b, 2013a) to freezing conditions and showed that there is a relationship between unfrozen water content and both porosity and cation exchange capacity (CEC). The authors argue that the CEC is a function of surface area and charge, which are parameters controlling the polarization of the Stern layer. To describe the quadrature
conductivity for freezing conditions, they used an exponential freezing function adapted by Duvillard et al. (2018). Such function provides a smooth transition between unfrozen and frozen states. In the model of Duvillard et al. (2018) and Coperey et al. (2019), they do not account for the high-frequency ice response observed in different laboratory analysis (e.g., Coperey et al., 2019; Duvillard et al., 2018; Kemna et al., 2014a). The polarization of pure ice (e.g., Auty and Cole, 1952) and ice-silicate mixtures (e.g., Stillman et al., 2010) has been characterized in the laboratory and analyzed at the field scale by Grimm
and Stillman (2015). Bittelli et al. (2004) developed a four-phase mixing model to determine volumetric liquid water and ice contents in a porous medium relying on the dielectric permittivity measured at two frequencies close to the relaxation frequency of ice. The characteristic dielectric relaxation of ice is due to the rotation of charge defects (Bjerrum L-defects) in the hydrogen-bonded ice lattice, which causes a reorientation of the permanent dipole of the $H_2O$ molecule. A temperature increase from -10°C to 0°C, in turn, increases the relaxation frequency of pure ice from 4 to 11 kHz (Auty and Cole, 1952), corresponding to
a resistivity phase peak that shifts from 26 to 71 kHz (Grimm and Stillman, 2015). Grimm and Stillman (2015) calculated the frequency effect (FE) (see, e.g., Vinegar and Waxman, 1984) as the fractional rate of change of $\rho'$ between two frequency endmembers from laboratory data and used power-law fits to determine ice volumes from the FE at each temperature. These laboratory FE relations together with SIP field data of a frozen silt permafrost site (Grimm and Stillman, 2015) could be used to obtain ice volumes for the study site. For silicate-ice mixtures, Stillman et al. (2010) found different polarization mechanisms
due to the interfacial water and soluble ions present between ice and silicates, further described in Bittelli et al. (2004).

## 3 Experimental setup and procedures

### 3.1 Site description

The study site Lapires is located in the Valais Alps (Western Swiss Alps, 2350-2700 m above sea level) and is characterized by a large northeast oriented talus slope (~500 m width). The slope consists of metamorphic blocks (mainly gneiss and schists)
reaching a thickness of more than 40 m (Scapozza, 2013), with finer-grained material in the Eastern part and coarse-blocky material in the Western part (see Fig.1). Meteorological data are monitored since 1998 with mean annual air temperatures around 0.5 °C (PERMOS, 2019). The permafrost distribution within the talus is discontinuous, and its thermal regime can be





characterized as temperate permafrost close to the melting point (Scapozza, 2013), and is influenced by an internal air circulation further described in Delaloye and Lambiel (2005) and Wicky and Hauck (2017). This so-called "chimney effect"
leads to ascending warm air in winter and descending cold air in summer and causes a net cooling of parts of the slope, as only cold atmospheric temperatures can convectively penetrate the coarse blocky surface, the latter having low thermal conductivities which effectively insulates the subsurface from high summer temperatures.

The composition of the talus slope was derived from four boreholes drilled between 1998 and 2008 (Scapozza, 2013), geophysical measurements (Delaloye, 2004; Hilbich, 2010; Lambiel, 2006; Mollaret et al., 2019) and ground temperature (GT)
records (Staub et al., 2015). According to borehole data, the ice-rich permafrost body reaches a thickness of up to 15 m and is overlain by a 5.2 m thick active layer (active layer thickness of the year 2019) (Scapozza et al., 2015a; Staub et al., 2015; ). A permanent ERT monitoring profile was installed in 2006 (Hilbich, 2010) to observe spatio-temporal dynamics of the freezing and thawing processes; the corresponding ERT results over more than a decade are described in Mollaret et al. (2019). Hilbich (2010) further compared seasonal changes in electrical resistivities with corresponding seismic travel times and borehole data.
Earlier results from the Lapires talus slope by Delaloye (2004), Lambiel (2006) and Staub et al. (2015) suggest that the spatial extent of ice-rich permafrost is most probable in a polygon-shaped zone, which is delineated in Fig. 1, and which we are referring to within this manuscript (cf. Fig. 1). Using borehole geophysics, Scapozza et al. (2015a) estimated the apparent porosity within the permafrost layer between 40 to 80 %, and a volumetric ice content between 20 to 60 %, with parts completely saturated with ice. Additionally, at the borehole locations, Marmy et al. (2016) (by thermal modelling) and Mollaret
et al. (2020) (by geophysical joint inversion) estimated the porosity to be around 40-60% and an ice content in the range of the mean borehole geophysical estimates.

### 3.2 SIP measurement setup

In our study, we present data collected in the frequency domain using the eight-channel DAS-1 system (from Multi-Phase Technologies) along five parallel horizontal profiles with roughly east-west direction and two vertical profiles (as depicted in
Fig.1). For consistency, we use the same profile notation as PERMOS with profiles H1 and V1 (not shown in this study) representing the long-term ERT monitoring profiles (Permos, 2019). The geophysical data presented in this study were acquired in two campaigns: (a) an initial survey conducted at the end of August 2018 to test different cable settings, and (b) a second survey at the beginning of September 2019 to map the extension of the permafrost. Both surveys were conducted during periods corresponding to well-developed thaw layers. Borehole temperature data were used for validation of the geophysical
results, including the temperature profiles of three boreholes where permafrost has been observed (BH-1108, BH-1208 and BH-0198) and one borehole where no permafrost was found during drilling (BH-1308) (see Fig. 1b).

### 3.2.1 Measurements to investigate data quality using different cables and layouts

An important focus of our study aimed at finding the field procedure that provides high-quality SIP imaging data sets with minimum disturbance by EM coupling, yet robust and easy to deploy in rough terrains, such as high-alpine permafrost



landforms. To this end, we conducted SIP measurements using two different cable types: multicore (mc) and coaxial cables (cc), as well as two different cable layouts: a single cable thread and separated cables. The latter corresponds to the use of two cables for each electrode position to minimize crosstalking between the cables, with one cable used to inject current and the second for voltage measurements. In particular, four different settings were compared: a) permanently installed single multicore cables along profile H1 (4 m electrode separation and 43 electrodes), b) single coaxial cables along profile H2 (32

electrodes with 5 m separation), c) separated multicore and d) separated coaxial cables (24 electrodes and 5 m separation) along a part of profile H2 (first electrode corresponds to the twelfth electrode of profile H2, cf. Fig. 1). The single coaxial cable and the separated cable set-ups were placed at a different locations (H2) to avoid noise in the data due to the presence of the permanently installed electrodes and cables from profile H1. The deployed coaxial cable consists of a bundle of independent coaxial cables with the shield of all wires attached at the end-connector to permit grounding through the measuring device (for

further details see Flores Orozco et al. (2021)). The ERT monitoring multicore cable was permanently installed in 2008 in the context of the PERMOS ERT network using common wires with no specific isolation between wires or other prevention of EM coupling, as this cable is only used for ERT monitoring. Multicore cables used in this study correspond to standard cables supplied by Iris Instruments. Data quality was evaluated by means of normal and reciprocal (N&R) analysis, with a reciprocal reading referring to a repetition of the measurement after interchanging current and potential dipoles. Accordingly, N&R data

were collected for each profile (cf. Table 1). Analyses of the misfit between N&R readings were used for the identification of outliers in the data (related to high discrepancies between N&R readings) and the quantification of data error (corresponding to small fluctuations between N&R), as described in previous studies (e.g., LaBrecque et al., 1996; Orozco et al., 2012; Slater et al., 2000).




**Figure 1: a) the Lapires study site located in the Western Swiss Alps, with information about the assumed spatial extent of the permafrost body according to Delaloye (2004), Lambiel (2006) and Scapozza et al. (2015a). SIP data were collected along 6 profiles, H1 denotes the PERMOS long-term ERT monitoring profile and H2 the cable test profile (see manuscript for further details). b) the borehole temperature data of BH-1108, BH-0198 and BH-1208 taken at the day of SIP measurements in August 2019. Note that BH-**
**1308 did not show permafrost during drilling (temperatures not shown). c) and d) show photographs of the talus slope, surface characteristics and the deployed coaxial cables. © Google Maps**



### 3.2.2 Mapping measurements to investigate the spatial distribution of permafrost

In August 2019, we collected SIP data along four additional profiles: H3, H4, V2 and V3 using 32 electrodes with a separation
of 10 m between them (Fig. 1). The position of the profiles was selected to cover ice-rich and ice-poor areas as well as different
substrates (e.g. fine-grained, blocky areas and bedrock). Contact resistances were measured before each data acquisition with
observed values between 5 and 60 kΩ, resulting in injected current intensities between 2-30 mA. SIP data were obtained
between 0.1 Hz and 225 Hz using a dipole-dipole (DD) configuration combining dipole lengths of 5 m (skip-0) and 20 m (skip-
3). The separation between current and potential dipoles extends over the entire length of the profile. We used stainless steel
electrodes; thus, the DD protocol was carefully designed to avoid collecting voltage readings with electrodes previously used
for current injection, which can result in erroneous SIP readings due to polarization of the electrodes (e.g., Flores et al., 2018).
The selection of the electrode separations (4 m and 5 m) and dipole lengths aimed at resolving the interface between active
layer and permafrost body, while measurements conducted with a larger electrode spacing of 10 m aimed at resolving the
permafrost base. The details of the acquisition are presented in Table 1.

**Table 1: Overview of the SIP measurement setup deployed along seven profiles to evaluate the data quality for different cable types
and set-ups and for mapping purposes.**

| profile name | orientation of profile | measure- ment date | number of electrodes | electrode spacing (m) | profile length (m) | cable type/setup | electrode configuration | number of quadrupoles | frequency range (Hz) |
|---|---|---|---|---|---|---|---|---|---|
| H1 | E-W | 09/2018 | 43 | 4 | 168 | single mc | DDsk3 N&R | 594 | 0.1-225 |
| H2 | E-W | 08/2019 | 32 | 5 | 155 | single cc | DDsk0,3 N&R | 435, 276 | 0.5-225 |
| H3 | E-W | 08/2019 | 32 | 10 | 310 | single cc | DDsk3 N&R | 276 | 0.5-225 |
| H4 | E-W | 08/2019 | 32 | 10 | 310 | single cc | DDsk3 N&R | 276 | 0.5-225 |
| V2 | N-S | 08/2019 | 32 | 10 | 310 | single cc | DDsk3 N&R | 276 | 0.5-225 |
| V3 | N-S | 08/2019 | 32 | 10 | 310 | single cc | DDsk3 N&R | 276 | 0.5-225 |
| H2 | E-W | 09/2018 | 2x24 | 5 | 155 | separated mc | DDsk0,3 N&R | 231, 120 | 0.1-225 |

### 3.3 Data processing and inversion of the SIP data sets

The N&R analysis is based on the concept that normal and reciprocal readings should be identical, with small variations
typically indicating the influence of random noise, while large changes point to readings affected by systematic errors, for
instance due to a poor galvanic contact in one of the electrodes or polarization of electrodes (e.g., Binley et al., 1995; Flores
Orozco et al., 2021; Huisman et al., 2016; Zimmermann et al., 2019). Accordingly, for each normal and reciprocal pair, we
can compute the average impedance magnitude $Z_{NR}$ and phase $\phi_{NR}$ as well as their corresponding misfit ($\Delta Z$ and $\Delta\phi$,
respectively). In case of random noise, we expect a normal distribution for $\Delta Z$ and $\Delta\phi$, respectively, which allows us to use
the standard deviation of the misfits $sd(\Delta Z)$ and $sd(\Delta\phi)$, respectively, to quantify the magnitude of the random noise
(LaBrecque et al., 1996) for each imaging data set (i.e., for each profile and each frequency). Moreover $sd(\Delta Z)$ and



$sd(\Delta\phi)$ can also be used for the identification of outliers. In particular for our study we defined as outliers ($Z_{f_{ilt}}$ and $\phi_{f_{ilt}}$) those quadrupoles for which:

$$Z_{f_{ilt}} = \frac{\Delta Z}{Z_{NR}} \geq 0.5 \quad \& \quad \Delta Z > 2sd(\Delta Z), \tag{4}$$

$$\phi_{f_{ilt}} = \frac{\Delta\phi}{\phi_{NR}} \geq 0.5 \quad \& \quad \Delta\phi > 2sd(\Delta\phi), \tag{5}$$

similar to the filtering routine described in Flores-Orozco et al. (2019). For the quantification of data error in the inversion, we used the well-accepted linear model describing the uncertainty in impedance magnitude ($s(Z_{NR})$) as a linear function of $Z_{NR}$, i.e.:

$$s(Z_{NR}) = a + bZ_{NR}, \tag{6}$$

with parameters $a$ and $b$ denoted as the absolute (in Ohm) and the relative (percental) error, respectively. As explained in detail by LaBrecque et al. (1996), such model gives the flexibility required to fit a wide range of impedance magnitude values. For the quantification of the $a$ and $b$ values, we used an approach similar to the one proposed by Slater and Binley (2006), where the absolute error $a$ ($\Omega$) and relative error $b$ (%) are given by:

$$a = \text{amean}(\Delta Z), \tag{7}$$

$$b = 100sd\left(\frac{\Delta Z}{Z_{NR}}\right). \tag{8}$$

For the quantification of data error in the impedance phase readings, we applied a constant- error model (i.e., no dependence on the impedance magnitude or phase values), obtained as $sd(\phi_{NR})$ similar to the approach by Slater and Binley (2006).

For the inversion of the SIP data, we used the finite-element smoothness-constraint inversion code CRTomo (Kemna, 2000), which uses impedance magnitude and phase values to compute the distribution of the complex resistivity in the subsurface for

every profile and frequency separately. This algorithm allows us to control the inversion of the data to a confidence interval defined by the data error estimates described by the error models in Equ. 7 and 8 (Kemna, 2000; Orozco et al., 2012). As demonstrated in several studies, this approach minimizes the risk of overfitting the data in the inversion and the generation of artefacts. We blanked those areas in the inverted images associated with cumulated sensitivity values two orders of magnitude smaller than the highest cumulated (absolute) sensitivity (normalized cumulative, error-weighted sensitivities), as described

by Weigand et al. (2017). We used the same error parameters ($a = 0.1\ \Omega$, $b = 5$ % and $sd(\phi_{NR})$) for the inversion of all SIP data sets to fit all data to the same error level enabling a better comparison of multi-dimensional (different profiles and frequencies) inversion results similar to Lesparre et al. (2017). In our analysis, we did not use a robust-inversion scheme (as proposed in the approach of LaBrecque and Ward (1990)) in which errors of data with large misfits are successively increased during the inversion.





## 4 Results

### 4.1 SIP data quality analysis for different cable setups

Figure 2 shows a modified version of the pseudosection, where we present both normal and reciprocal readings, in terms of the impedance magnitude and phase, after removal of outliers following the procedure described above. Pseudosections allow for an easy visualization of the spatial consistency in the readings and the position of the removed quadrupoles. In Fig. 2 we

compare the filtered pseudosections for data collected with single multicore cables, separated multicore cables, separated coaxial cables and single coaxial cables. Pseudosections are presented for impedance magnitude measurements at 0.5 Hz as we observed no frequency-dependence for the impedance magnitude (data not shown); while impedance phase readings are presented at four frequencies in the range between 0.5 and 75 Hz.

In general, the comparison of the four settings indicates that (a) measurements with coaxial cables show a higher data quality

(i.e. a lower standard deviation of $\Delta\phi$) than multicore cables and that (b), independent of the cables, the pseudosections become less smooth with increasing acquisition frequency and separation between current and potential dipoles (i.e. deep levels or large pseudodepths) due to a weaker consistency between the readings. All cable setups reveal an increase in the impedance phase with increasing frequency, with values between -40 and 0 mrad for the lowest frequency investigated (0.5 Hz); while at 75 Hz, the values range between -100 and -40 mrad. Single multicore cables show a high number of N&R outliers for all

frequencies (with less than 91 of 594 N&R pairs that remain after filtering) with a poor spatial consistency in the readings observed in the pseudosection. At low frequencies (0.5 Hz), the data quality is improved by separating the multicore cables, whereas at higher frequencies we observe higher standard deviations of the misfits. For instance at frequencies equal or above 7.5 Hz, less than 17 of 351 N&R pairs remain after filtering for separated multicore cables. In contrast, measurements collected with a single coaxial cable show smooth pseudosections (< 210 of 711 N&R pairs after filtering) down to a pseudodepth of

~40m indicating a significant improvement of the signal-to-noise ratio (SNR) compared to the other cable setups. Even for 75 Hz, there are still 172 of 711 N&R pairs remaining for single coaxial cables after filtering, whereas for separated multicore cables only 7 of 351 N&R pairs remained after filtering. When comparing single and separated coaxial cables, the data quality decreases for separated cables at a frequency of 75 Hz with only 44 of 351 N&R pairs kept after our N&R analysis. The observation of a higher number of outliers for separated cables when compared to single cables for higher frequencies is

unexpected, considering that separation of current and voltage cables is a standard technique used in IP surveys to decrease EM coupling, in particular cross-talking (see Dahlin et al., 2002). Our explanation for the poor quality of data collected with separated cables refers to the large variations in the contact resistances, considering that in this case we not only interchange the dipoles, but physically deploy different electrodes for the normal and reciprocal readings. Moreover, measurements with separated coaxial cables were collected without grounding the coaxial shields, likely resulting in cross-talking between the

shield and the conducting copper wire.





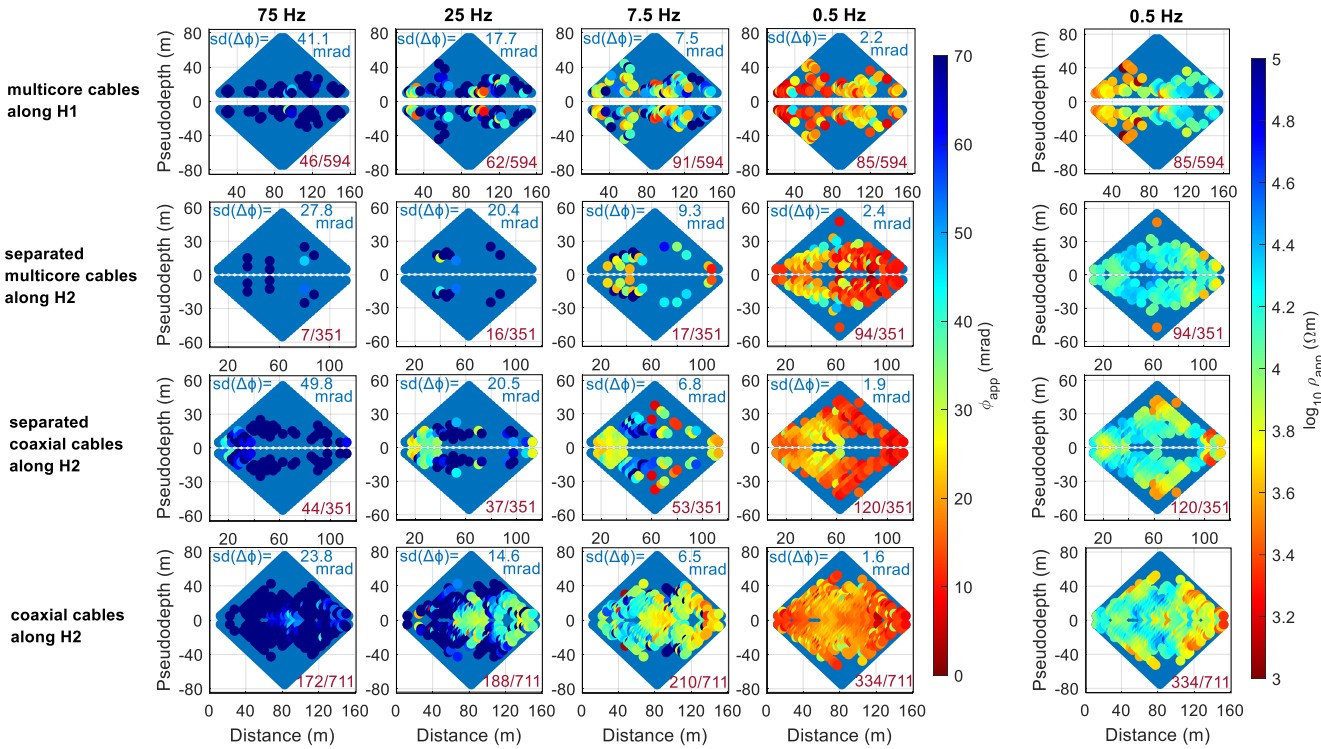

**Figure 2: Pseudosections for normal and reciprocal data at 4 different frequencies (0.5, 7.5, 25 and 75 Hz) collected at the Lapires site along the PERMOS monitoring profile H1 (green line in Figure 1, 43 electrodes with 4m spacing), along profile H2 (32 electrodes, 5m spacing) and along the orange part of profile H2 (24 electrode, 5m spacing). Pseudosections are presented before (blue symbols) and after (coloured symbols) the removal of outliers based on the misfit between normal and reciprocal readings (following Equ. 22 and 23) indicated by a label at the right bottom of each subplot referencing to the total and remaining number of N&R pairs. The standard deviation of the N&R misfit ($sd(\Delta\phi)$) is written in each plot for each frequency and cable setup.**

In Fig. 3, we present a statistical analysis of the misfit of normal and reciprocal phase $\Delta\phi$ and resistance $\Delta Z$ readings to quantitatively evaluate data uncertainty of the different cable setups. The histograms in Fig. 3 reveal, in general, a normal distribution, demonstrating that the filtered data sets do not contain significant systematic errors and are mainly affected by random noise. In general, the histograms reveal a larger variance in the N&R misfits for measurements collected with multicore and separated cables, demonstrating a significant improvement in the quality of the SIP data collected in permafrost terrain when using coaxial cables. We observe the smallest misfit values and smaller $sd(\Delta Z)$ and $sd(\Delta\phi)$ for single coaxial cables for all frequencies with an approximately 17 mrad smaller standard deviation and ~16 % fewer outliers in comparison to single multicore cables for 75 Hz. Even at low frequencies (e.g. 0.5 Hz), the enhanced S/N ratio for single coaxial cables is demonstrated by a ~0.6 mrad (25 %) smaller standard deviation and ~33 % fewer outliers removed compared to single multicore cables. For separated cables, we observe no further improvements in the data quality compared to single coaxial



cables. Hence, in our study we used a single coaxial cable as they permit easier field procedures than separated cables and reveal the best reciprocity, as observed in Figs. 2 and 3.

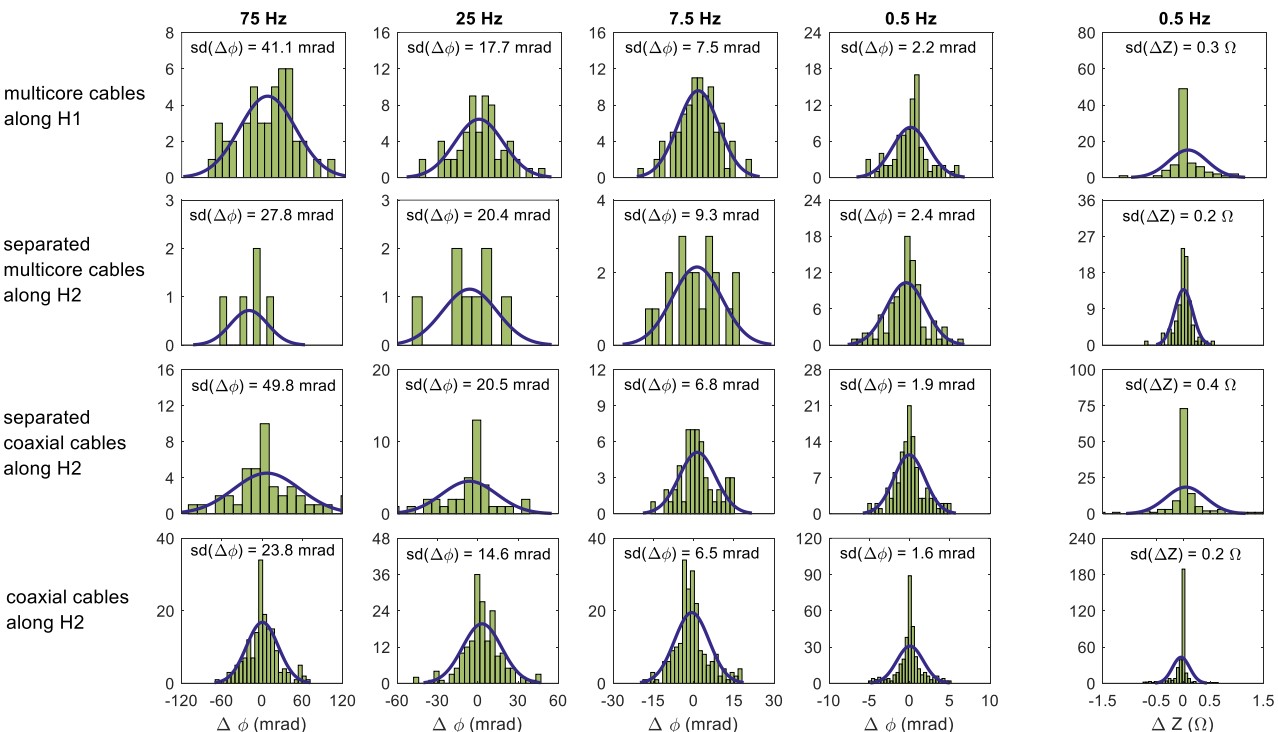

**Figure 3: Misfits in phase values between normal and reciprocal phase and resistance readings collected at different frequencies**
**(0.5, 7.5, 25, 75 Hz) and with different cable settings. The standard deviation of the N&R misfit ($sd(\Delta Z)$ and $sd(\Delta\phi)$, respectively) is given in each subplot for each frequency and cable setup representing a quantitative way to compare the different cable setups for all frequencies.**

### 4.2 Electrical resistivity imaging results: a first discrimination between ice-rich and unfrozen talus

We present in Fig. 4 resistivity imaging results for data collected along profiles H2, H3, H4, V2 and V3, which clearly reveal two main units, i.e., a low and a high resistivity unit. These resistivity variations coincide with the temperature records in the four boreholes, i.e. frozen conditions are represented by high resistivities and vice versa. The borehole positions are indicated in the resistivity images including thaw depth and permafrost base at the date of the measurement. By extracting specific resistivity values from the inversion result for all profiles at close proximity to a borehole at the depth of the permafrost base,

a mean value for the boundary between ice-rich and unfrozen talus of 10,000 Ωm can be estimated (see dashed line in Fig. 4). The depth of the active layer cannot be identified from the vertical resistivity contrast, as the electrode separation used in the survey was too large to resolve small structures with a smoothness-constraint inversion approach.





Fig. 4b shows the model parameters extracted at the intersection of two crossing profiles revealing a consistent vertical distribution of electrical resistivity values. The active layer is partly visible for H2 due to the smaller electrode spacing of 5 m,

but is still hard to distinguish from the ice-rich permafrost because of similar electrical resistivity values for air- and ice-filled talus. In general, all profiles reveal resistivity values in the range of 10 to 100 kΩm corresponding to the porous, coarse blocky, and ice-rich permafrost layer, and a clear contrast to unfrozen, fine-grained parts with unconsolidated blocks ($1 < \rho < 10$ kΩm) and the unfrozen till below the permafrost base ($1 < \rho < 10$ kΩm). These results are consistent with previous investigations at the Lapires talus slope (Hilbich, 2010; Mollaret et al., 2019, 2020), who report a clear resistivity and seismic velocity contrast

between the host material of the talus slope (unconsolidated blocks and fine-grained material) and the presence of ice-rich permafrost.

The right-hand part of profile H3 is characterised by the transition from the talus slope into a bedrock slope with a thin sediment/vegetation cover (250 – 310 m horizontal distance). The observed resistivity range of the bedrock slope is with 10-29 kΩm comparable to the resistivity of the ice-rich talus (110 – 240 m horizontal distance), which demonstrates the general

ambiguity and the challenges of a direct interpretation of the resistivity values (as frozen or unfrozen bedrock, or air/blocks) in the absence of ground truth. In this context, SIP data may help to improve the discrimination between ice-rich talus slope and bedrock based on their characteristic spectral behaviour, which will be further discussed below (Discussion section).



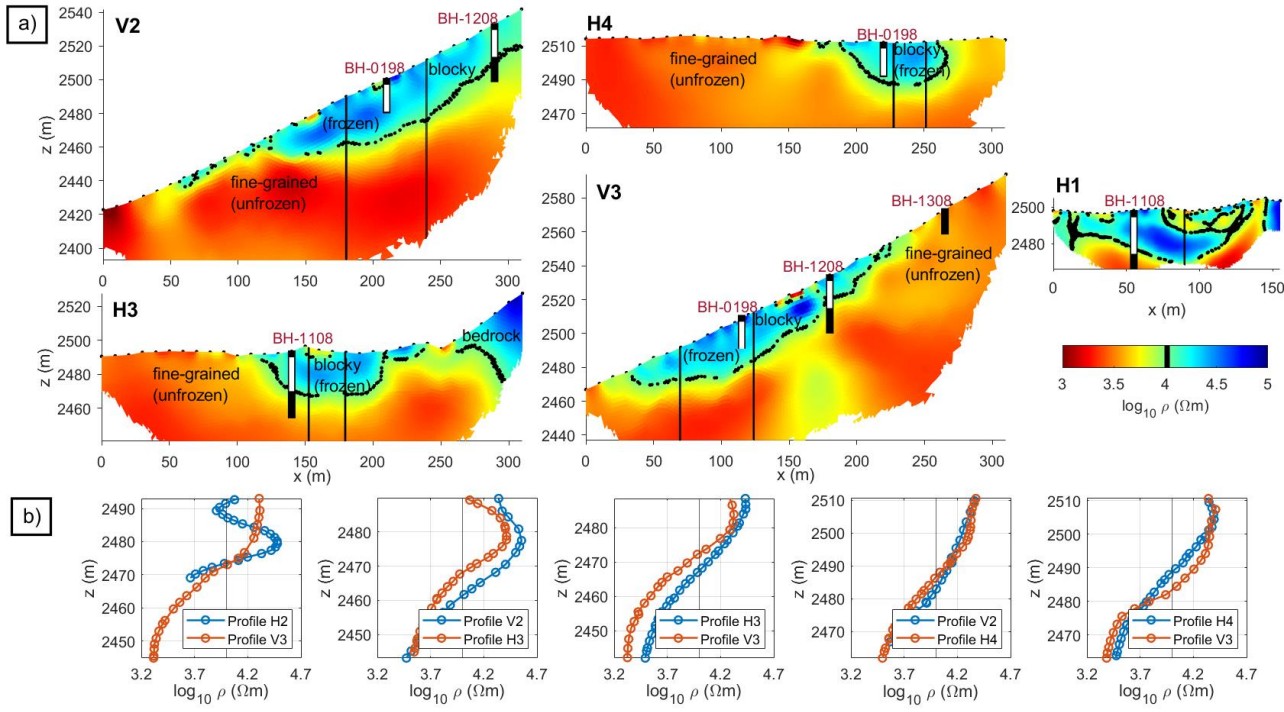

**Figure 4: a) Resistivity imaging results and their interpretation. The transition between frozen and unfrozen subsurface materials at ~ 10 kΩm is indicated by the dashed black line, and the boreholes along the resistivity profiles highlight the thaw depth and permafrost base at the date of the measurement, respectively. b) Comparison of the vertical profiles of the electrical resistivity values extracted from the resistivity images at the intersection between crossing profiles.**

### 4.3 Frequency-dependence of the measured impedances of ice-rich and unfrozen talus

Before discussing the induced polarization imaging results, it is recommended to look at the frequency dependence in the electrical impedances, expressed in terms of the apparent resistivity ($\rho_a$) and phase shift ($\phi_a$). Fig. 5 shows the spectra collected along profile V2 for the 2nd and 3rd level (i.e., for 20 m and 30 m separation between the current and potential dipoles) from north (downslope) to south (upslope), which corresponds to a transition from unfrozen to frozen materials. As expected for the investigated frequency range, we do not observe a frequency-dependence for $\rho_a$; however, the values differ by one order of magnitude between the unfrozen and frozen part of the talus slope (Fig. 5a, c). Likewise, for the apparent phase shift of the electrical impedance, a distinction between frozen and unfrozen states seems to be possible with higher absolute $\phi_a$ values observed for ice-rich parts (Fig. 5b), although SIP data are partly obscured due to the low SNR. Additionally, Fig. 5d shows a general increase in the absolute $\phi_a$ values with increasing acquisition frequency. However, such increase is with ~50 to ~360 mrads between 7.5 and 225 Hz more pronounced in the ice-rich part of the talus slope (> 145 m horizontal distance), compared to the unfrozen zone (< 145 m horizontal distance, ~25 to ~60 mrads). This increase in the absolute $\phi_a$ values at high



frequencies (> 7.5 Hz) in the areas where frozen material is expected is consistently observed also in the other profiles, and highlights the different response for frozen and unfrozen material.

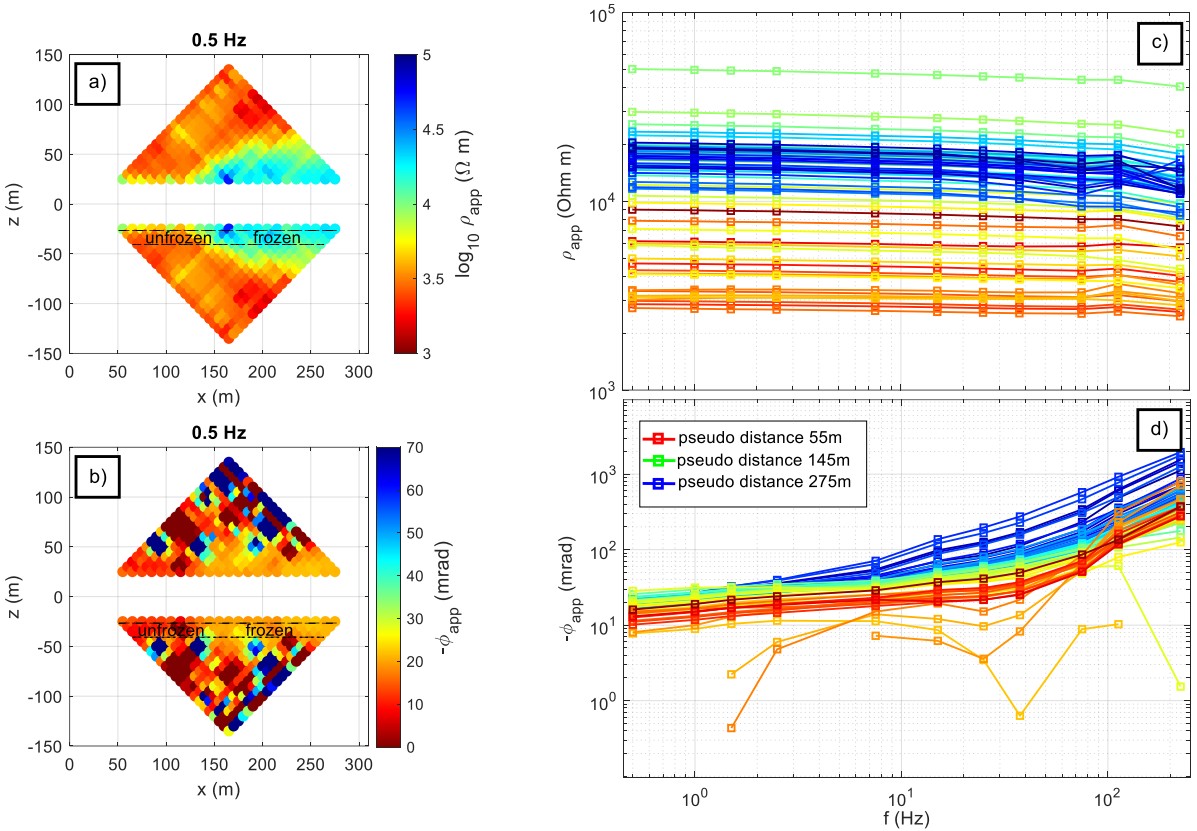

**Figure 5: Visualization of the frequency-dependence in the electrical impedance measurements collected along profile V2. Impedance magnitude values are converted to apparent resistivity (a) and phase (b) values. The spectral response (c, d) is plotted for level 2 and 3 (i.e. 20 and 30 m electrode separation between current and potential dipoles, between the dashed lines) from north (downslope) to south (upslope) corresponding to the transition from unfrozen to frozen subsurface conditions.**

### 4.4 SIP imaging results: differences in the complex resistivity permit an improved interpretation of permafrost

To investigate variations in the polarization response, we present in Fig. 6 exemplary imaging results expressed in terms of the phase ($\phi$) as well as the real ($\rho'$) and imaginary ($\rho''$) components of the complex resistivity (CR), obtained for data collected along profiles H3 (Fig. 6a) and V3 (Fig. 6b) for low (0.5 Hz) and high (75 Hz) frequencies. In general, both profiles reveal high polarization, expressed by higher absolute $\rho''$ and $\phi$ values, for frozen, ice-rich and blocky areas of the talus slope; while lower values are related to unfrozen, fine-grained parts of the talus. Additionally, we observe generally higher absolute $\rho''$ and $\phi$ values for 75 Hz than for images at 0.5 Hz, which is expected considering the spectra of the measured impedances (c.f. Fig.



5). Similar to Fig. 4, an interpretation of the frozen and unfrozen parts of the talus slope is indicated in the images of $\rho'$ and $\rho''$, as presented in Table 2. Phase images do not show a clear correlation with the borehole data.


**Table 2: Range of complex resistivity values observed in the inversion results of all profiles for different subsurface materials**

| | $\rho'$ ($\Omega$m) (0.5 Hz) | $-\rho''$ ($\Omega$m) (0.5 Hz) | $-\varphi$ (mrad) (0.5 Hz) | $\rho'$ ($\Omega$m) (75 Hz) | $-\rho''$ ($\Omega$m) (75 Hz) | $-\phi$ (mrad) (75 Hz) |
|---|---|---|---|---|---|---|
| fine-grained unfrozen | 2,400-10,000 | 1-160 | 1-20 | 2,000-10,000 | 90-1,260 | 40-60 |
| blocky frozen | 10,000-29,000 | 160-970 | 20-40 | 10,000-29,000 | 1,260-3,000 | 60-200 |
| bedrock | 5,000-15,500 | 50-160 | 1-25 | 5,000-15,500 | 150-1,260 | 30-60 |

The first 120 m of profile H3 (Fig. 6a) reveal in general low values for all parameters (e.g. Table 2) at 0.5 and 75 Hz related to the fine-grained, unfrozen material. The middle part of H3 shows much higher values than the unfrozen area for both 0.5

and 75 Hz, clearly delineating lateral variations in the subsurface, which we interpret as the blocky, ice-rich material of the talus slope. The right part of profile H3 covers the bedrock slope (> 250 m horizontal distance) and shows intermediate values, which differ from both the unfrozen and the ice-rich talus material. While the imaging results for the polarization (both $\rho''$ and $\phi$) reveal a clear anomaly with high absolute values between ca. 120 and 200 m horizontal distance (the extent of the frozen, ice-rich material), the resistive anomaly between 250 and 300 m horizontal distance shows much lower polarization

values (in particular at 75 Hz), which allows to better discriminate between ice-rich talus and (unfrozen) bedrock.



**Figure 6: Complex resistivity inversion results in terms of real and imaginary components and phase at 75 Hz and 0.5 Hz for profiles a) H3 and b) V3. Our interpretation of frozen and unfrozen parts of the talus slope is indicated by the dashed line, corresponding to**
**values of $\rho' = 10,000$ Ωm, $\rho'' = 160$ Ωm (at 0.5 Hz) and 1,260 Ωm (at 75 Hz). The boreholes BH-0198, BH1108, BH-1208 and BH-1308 indicate the thaw depth, permafrost body and permafrost base at the date of the measurement.**

Profile V3 in Fig. 6b also demonstrates the improved interpretation of subsurface properties by means of induced polarization $(\rho', \rho'', \phi)$ over the interpretation of only resistivity images (e.g., from ERT). From the calibration at the location of the two
permafrost boreholes (BH-1108 and BH-1208) it can be concluded that absolute $\rho''$ values exceeding 160 Ωm for 0.5 Hz and 1,260 Ωm for 75 Hz can be interpreted as permafrost (cf. Table 2). The interpretation of the inversion results for the unfrozen part (220-310 m) resolved in profile V3 is further validated by comparison with the non-permafrost borehole BH1308, showing positive temperatures and an absence of ground ice observed during drilling (cf. PERMOS 2019). Extending this permafrost




criterion for $\rho''$ to the entire inverted image, it can be seen that the lateral extent of the permafrost body characterized by $\rho''$
is slightly smaller for 75 Hz compared to 0.5 Hz, and compared to $\rho'$. Additionally, we can observe the clearest contrast
between the blocky, ice-rich material of the talus slope and the bedrock slope for $\rho''$ at 75 Hz, suggesting an even higher
sensitivity at this frequency to discriminate ice-rich locations. According to this interpretation $\rho''$ would mainly differ for ice-
filled pores and pores without ice, whereas $\rho'$ cannot clearly discriminate between ice- and air-filled pores; however, no further
ground truth data is available to validate this interpretation.


To investigate in detail the frequency dependence of the electrical properties, we show in Fig. 7 CR values extracted from a
virtual borehole in a) the unfrozen part of profile H4 and b) the ice-rich permafrost zone of profile V2. The values were
extracted along the vertical black line (mean value over 2 m pixel width) for inversion results independently obtained at each
frequency (0.5 - 75 Hz). The different spectral behaviour allows a clear distinction between frozen and unfrozen conditions.
For the unfrozen part in profile H4 we observe low real and absolute imaginary resistivities ($\rho' \sim 3000$ Ωm, $\rho''$ between 20 -
300 Ωm) and a weak frequency dependence in the investigated range. In contrast, the ice-rich part in profile V2 shows
substantially higher resistivity and polarization values ($\rho' \sim 15{,}000 – 30{,}000$ Ωm, $-\rho''$ 300 – 4,000 Ωm) and a pronounced
frequency dependence in the response, with absolute $\rho''$ values increasing with increasing frequency in the uppermost 20
metres compared to deeper parts. These shallow 20 m correspond to the vertical permafrost extent within the talus slope, as
observed in the borehole temperature data. Yet, we are not able to resolve the extension of the active layer due to the relatively
large electrode spacing chosen for the acquisition of the data, an issue which will be addressed in the next section.

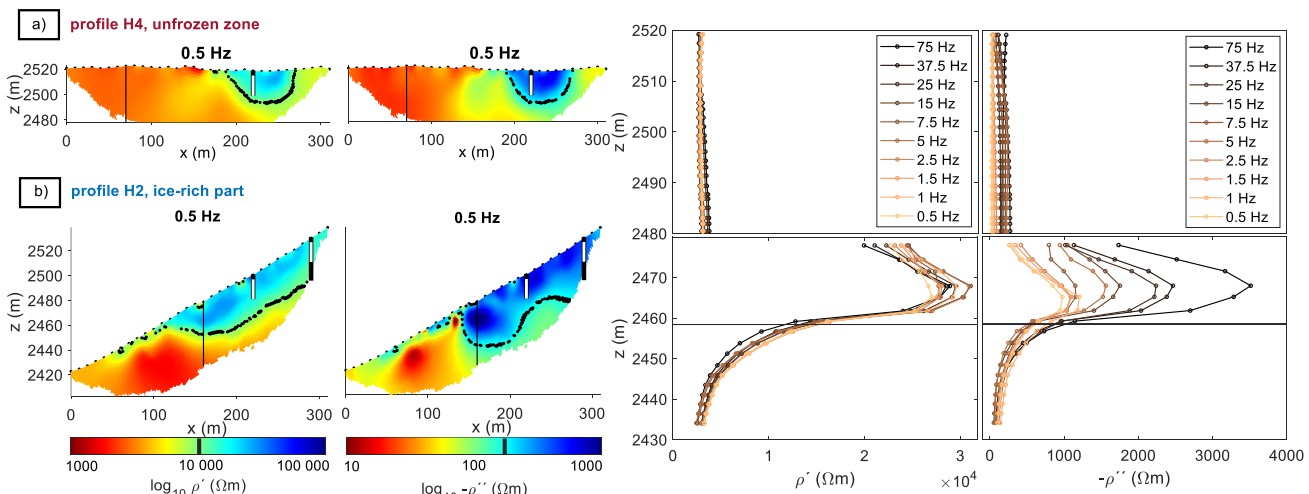

**Figure 7: Extracted complex resistivity values (real and imaginary components) in a virtual borehole in an unfrozen zone (top,**
**profile H4) and in an ice-rich part (bottom, profile V2) of the study area indicating the different spectral responses for the two**
**positions. Extracted values correspond to mean values over a pixel width of 2 m along the vertical black lines in the tomograms.**



## 5 Discussion

### 5.1 Reliability of SIP data

In complex resistivity imaging data, EM coupling is known as an important source of error affecting $\phi$ readings especially at frequencies > 10 Hz (e.g., Kemna et al., 2012). Nonetheless, the study of Flores Orozco et al. (2021) shows EM coupling affecting SIP readings collected with common multicore cables already at low frequencies such as 1 Hz, while the use of coaxial cables permitted to collect data with high quality at much higher frequencies in conductive soils. As demonstrated in Fig. 2 and Fig. 3, the deployment of coaxial cables also improves the SIP readings in high resistive environments, as in the

present case of the Lapires talus slope. The data error in measurements collected with multicore cables is higher (cf. Fig. 3), even if separate cables are deployed for current and potential dipoles. The latter is the standard procedure for the collection of reliable time-domain IP data, and Duvillard et al. (2018) demonstrated its applicability for alpine permafrost. However, in our analysis the use of separated cables revealed poor N&R misfits. In our study we used two parallel lines of 24 electrodes (current and potential dipoles each) with common multicore cables separated by ca. 0.5 m. The readings revealed a poor reciprocity

above and at 7.5 Hz as observed in Fig. 2 and Fig. 3, which is not consistent with previous studies (Dahlin et al., 2002; Duvillard et al., 2018; Flores Orozco et al., 2021). Although further research might be necessary to understand this issue, we believe the problem is caused by the use of two different electrodes to separate current and potential measurements at the same position. The potentially large variation in the respective contact resistances as a consequence of the coarse-blocky surface of the talus slope may cause the poor reciprocity observed. An alternative to conduct measurements with separated cables could be to

assign, for instance, current injections with odd electrode numbers and potential readings with even numbers. In such case, we can still separate current and potential dipoles with different cables, still collect data over the same profile length as a single cable layout, but avoid having two electrodes for the same position. Such approach would not allow for skip-0 protocols and, thus, in this study we opted for the layout with two separated cables and electrodes to permit data collection with exactly the same electrode configuration and electrode spacing.

Based on the use of single coaxial cables, our analysis demonstrates the possibility to collect data with a high reciprocity in the range between 0.5 and 75 Hz, with a higher reciprocity in the higher frequencies compared to the study of Flores Orozco et al. (2021). This is likely due to the high resistivity of the substrate which shifts inductive coupling to higher frequencies. Accordingly, our data reveal poor reciprocity for readings collected above 100 Hz resulting in just a small number of quadrupoles remaining after filtering. Thus, we reduced our interpretation of the imaging results to data collected at 75 Hz and

below. Although several quadrupoles had to be discarded above 100 Hz, some of the measurements collected with small separations between current and potential dipoles show a high SNR and smooth impedance phase spectra for the entire frequency range as observed in Fig. 5. Only by looking at the frequency-dependence of the electrical impedance, a distinction between frozen and unfrozen conditions can be found that is consistent with nearby borehole temperature data (cf. Fig. 1). In well-controlled conditions in a permafrost tunnel of frozen silts, Grimm and Stillman (2015) collected data with the SIP Fuchs

III instrument (from Radic research) with a 20 kHz bandwidth, which permits the digitalization of the data in remote units



directly connected with the electrode, effectively minimizing EM coupling at frequencies below 100 Hz (e.g., Martin et al., 2020) and inductive coupling at frequencies up to 20 kHz (e.g., Grimm and Stillman, 2015). However, the deployment of such a system dramatically increases the field work in comparison to the laying of a single cable. Thus, for our study and further investigations in alpine permafrost, we believe that the use of a single coaxial cable might be the best compromise considering
the improved data quality (i.e., reciprocity) and simple procedures for field surveys.

The consistent analysis of the SIP data collected in different profiles and at multiple frequencies revealed the possibility to quantitatively correlate spatial variations in the complex resistivity with variations in the ice content and geological units (Fig. 8). As discussed in Kemna (2000) and Flores Orozco et al. (2011; 2012), the use of error models in the inversion of the data and a careful quantification of the data error permit to fit the different data sets to adequate confidence intervals and to correctly
retrieve the frequency dependence of the electrical parameters during the inversion. Grimm and Stillman (2015) used the time-lapse feature of RES2DINV to invert the entire frequency spectrum, which applies a regularization across successive measurements but without adequate error parametrization in the inversion. In our analysis, we took special care in the quantification of the data error, but performed an independent inversion of the data sets for all frequencies, accounting only for spatial regularization across each imaging plane. An alternative approach (e.g., Günther and Martin, 2016; Kemna et al.,
2014b; Son et al., 2007) relies on a simultaneous inversion of SIP data measured over a range of frequencies. Although multi-frequency inversion may improve the consistency between results and its interpretation, previous studies have demonstrated that adequate error quantification also permits a quantitative interpretation of SIP imaging results even without any spectral regularization (e.g., Flores Orozco et al., 2013). The consistency between imaging results obtained for different profiles (cf. Fig. 4b), at different frequencies and with borehole data demonstrates the possibility to rely on this approach.

**5.2 Spatial patterns of the SIP response and its applicability to delineate the spatial permafrost extent**

With this measurement setup for alpine permafrost and based on the proposed methodology to process and invert the SIP data sets, we present in Fig. 8 a synopsis of all profiles for a characterization of the spatial extent of permafrost at the Lapires test site. The complex resistivity at 5 Hz is shown in terms of the real (Fig.8a) and imaginary (Fig. 8b) components with additional spectra depicted at selected locations (Fig. 8c). The values are shown as arithmetic mean at a depth of 10 meters at each
electrode position. The interpreted permafrost extent using the thresholds defined in the Results section is indicated for three different depths (7 m, 10 m, 15 m). A spline function was fitted to obtain three polygons, which can be compared to the assumed permafrost distribution from previous studies. When comparing the polygons delineating the permafrost extent at different depths for the real and imaginary components, we observe a reduction in the size with depth and an overall smaller spatial extent of permafrost compared to the original polygon (cf. Fig. 1). The geometry of this polygon was established by
Delaloye (2004) between 1998 and 2004 using bottom temperature of snow cover (BTS), universal temperature logger (UTL), vertical electrical sounding (VES) and borehole temperature measurements. Thus, caution must be taken when comparing our results to the study of Delaloye (2004), as we used a different methodology, equipment and resolution. Nevertheless, the reduction in permafrost extent that is suggested by the results of this study is consistent with the findings of Mollaret et al.





(2019), who also report a strong and spatially consistent resistivity decrease over the whole monitoring profile H1, and hence
a decrease in ice content at the Lapires talus slope (based on long-term resistivity changes over one decade from 2006 to 2017). Mollaret et al. (2019) state as well that amongst all ERT-monitored permafrost sites in the Swiss Alps, the Lapires talus slope shows one of the most severe increases of permafrost temperature. A thorough analysis whether the observed reduction in permafrost extent within our study compared to the estimate by Delaloye (2004) is a result of permafrost degradation or solely based on methodological differences is, however, beyond the scope of this study.

Real and imaginary resistivity at 0.5 Hz yield similar results for the extent of permafrost with slightly smaller polygons delineated for $\rho''$. Duvillard et al. (2020) used conductivity and induced polarization to infer the temperature distribution of a permafrost-affected rock ridge and postulate that it only supports ERT data in assessing the contribution of the surface conductivity to the total conductivity of the material. However, they did not investigate the frequency dependence of the IP, as TDIP field data were acquired at a dominant frequency of ~1 Hz. Since Kemna et al. (2014a), Wu et al. (2017), and Coperey
et al. (2019) report an increase in the polarization response at frequencies above 10 Hz with increasing ice content, in Fig. 8c we present complex resistivity spectra in a frequency range between 0.5 and 225 Hz that reveal a similar pattern to those studies.



**Figure 8: Characterization of the spatial extent of permafrost of the Lapires talus slope for the complex resistivity expressed as real $\rho'$(a) and imaginary $\rho''$(b) components at 0.5 Hz with additional spectra depicted at selected locations (c). The complex resistivity values represent the arithmetic mean at a depth of 10 meters at each electrode position. Based on the threshold values defined in Fig. 6, the permafrost extent was estimated for each profile for three different depths (7 m, 10 m and 15 m) and a spline function was fitted to obtain the white, grey and black dashed lines, which can be compared to the blue polygon (cf. Fig. 1). The spectral responses are shown for different locations (coloured squares) in the talus slope (see legend for description). © Google Maps**

In Fig. 8c, we depict the spectra of the complex resistivity for different substrates and thermal conditions: unfrozen, fine-grained talus; frozen, blocky talus and bedrock. Here, pixel values represent the mean of $\rho'$, $\rho''$ and $\phi$ of the CR extracted from the area between 5 and 15 m depth and with a horizontal width of 2 m. The plots reveal clear variations in the electrical parameters, but larger variations in the shape of the polarization spectra, i.e., in the frequency dependence of $\rho''$ and $\phi$, than for $\rho'$ between frozen and unfrozen states and between different substrates. For the fine-grained, unfrozen material spectra, we observe a consistent trend with the lowest $\rho'$, $\rho''$ and $\phi$ values compared to other substrates and thermal conditions. The





polarization effect (expressed in terms of $\rho''$ and $\phi$) exhibits low (absolute) values for lower frequencies and a rapid linear increase with frequency above 37.5 Hz. For the frozen and blocky subsurface materials, the observed values of $\rho'$ and $\rho''$ are

higher by an order of magnitude compared to the unfrozen materials. The spectral behaviour is different for frozen, blocky material, with increasing $\rho''$ and $\phi$ values with increasing frequency from 7.5 Hz onwards. The $\rho^*$ ($\rho'$, $\rho''$ and $\phi$) values for bedrock are moderate, in between those of the other substrates.

As already depicted in the imaging results of Fig. 6 and observed in other permafrost studies using only electrical resistivity tomography (e.g., Mollaret et al., 2020; Steiner et al., 2021), the interpretation of $\rho'$ alone is not always conclusive.

Accordingly, there is still considerable uncertainty regarding the thermal condition of the bedrock in Lapires, as neither boreholes, nor geophysical profiles are available that confirm frozen or unfrozen conditions in the bedrock part of the site. To discriminate between frozen and unfrozen bedrock, both the four-phase model and petrophysical joint inversion reach their limits and require additional information (e.g., Hauck et al., 2011; Wagner et al., 2019). Our results suggest that SIP data may provide such extra information to distinguish between different units in ambiguous cases, by identifying unfrozen bedrock

based on the local maximum at low frequencies (~10 Hz for $\phi$) in both $\rho''$ and $\phi$ images (cf. Fig. 8c). As shown for sandstones, the induced polarization relaxation characteristics are governed by the dynamic pore radius (e.g. Revil et al., 2012b). To characterize the textural and mineralogical controls on the SIP behaviour of the bedrock in Lapires, we collected solid rock samples at the site and Limbrock et al. (2021) analysed the temperature dependence of its polarization response in the laboratory. The sample represents an igneous rock (e.g. granite) sample with a porosity of 2% consisting of 39% muscovite,

36% quartz, 17% albite 4% microcline and 4% clinochlore. Limbrock et al (2021) used a Debye decomposition and calculated the relaxation time distribution of the unfrozen rock sample and compared it to the pore radius distribution. They found a peak relaxation time at around 10 ms and a peak pore size at around 200 nm. The corresponding peak frequency of ca. 15 Hz is consistent with the local maximum of the exemplary bedrock $\phi$ spectra of profile H3 in Fig. 8c. This finding supports our assumption that the bedrock might be unfrozen and that we are able to distinguish between ice-rich talus material and unfrozen

bedrock due to their distinct spectral behaviour. Additionally, the predominant occurrence of frozen conditions in coarse blocky terrain, compared to fine-grained material and bedrock under the same climatic input conditions, is well supported by experimental and modelling studies on the influence of surface and subsurface heterogeneity on borehole temperatures in mountain terrain (e.g. Schneider et al., 2012). These studies show that in a coarse blocky surface cover air convection processes take place and lead to a pronounced ground cooling, whereas thermal conduction is higher in bare bedrock sites leading to a

higher heat storage capacity and warmer subsurface temperatures (Wicky and Hauck, 2017, 2020).

**5.3 SIP response of active layer and permafrost body**

If complex resistivity values are markedly different for laterally varying unfrozen and frozen subsurface conditions as demonstrated above, they should show the same effect in vertical direction regarding the transition from the unfrozen active

layer to the permafrost body. In Fig. 9 we present the spectral response for pixel values extracted from the inversion results of



profile H2 close to borehole BH-1108, where the 5 m electrode spacing allows us to investigate shallow structures. The pixel values represent the mean values for a pixel width of 1 m for two different depth ranges: a) 0 - 5.6 m (thaw layer), and b) 5.6 - 19.5 m (permafrost). Previous geophysical investigations in close proximity to BH-1108 report low seismic P-wave velocities of 500-1500 m/s within the uppermost 4-5 m thick layer, and high electrical resistivity (10,000-100,000 $\Omega$m) and intermediate

seismic velocities (3,000-4,000 m/s) for the ice-rich permafrost zone below (Hilbich, 2010; Mollaret et al., 2020). Borehole analysis of BH-1108 by Scapozza et al. (2015a) and petrophysical joint inversion results of nearby seismic and ERT data (e.g., Mollaret et al., 2020) yielded mean ice contents across the entire image plane of 30 % with a thin supersaturated ice layer located at a depth of 15 m. Scapozza et al., (2015a) additionally reported four main layers in borehole BH-1108, listed from top to bottom: (1) a porous surface layer (active layer) with large air-filled voids (< 5 m depth), (2) a permafrost layer partially

saturated with ice (5 - 19 m depth), (3) a silty sand layer with low air contents (19 - 20 m depth), and (4) a till layer with larger air-filled voids (> 20 m depth). Temperature data of BH-1108 at the time of our SIP survey (Fig. 9a) allow us to determine a thaw layer thickness of 5.6 m and negative temperatures to a depth of 19.5 m, with unfrozen materials (positive temperatures) at larger depths.

Coperey et al. (2019) recently presented complex conductivity spectra for data collected in the laboratory in a broad

temperature range (between -15 °C and +20 °C), demonstrating the variations in the spectra for frozen and unfrozen materials (i.e. for saturated sandstone, granite, soil and sands). To permit a comparison with the results by Coperey et al. (2019), we plot in Fig. 9b the complex conductivity spectra retrieved for the active layer and the permafrost materials. Consistent with their laboratory study results, complex conductivity spectra in our field study exhibit a decrease of the real and imaginary parts of the complex conductivity with decreasing temperature. Similar to the study of Coperey et al. (2019), we observe an increase

of the imaginary component for the permafrost layer occurring at frequencies above 10 Hz, which we believe could be related to the dielectric relaxation of ice associated with Bjerrum defects as suggested by Stillman et al. (2010). In this regard, Wu et al. (2017) and report a shift of the phase spectrum during the freezing process of saline soil with an increase in the phase magnitude for frequencies > 100 Hz and a shift of the peak phase frequency to lower values across the whole spectrum. A similar behaviour was also described in Kemna et al. (2014a) with an interpretation in terms of membrane (electrochemical)

and ice polarization. In their study of rock permafrost, Grimm and Stillman (2015) found a temperature-dependent relationship between ice volume fraction and the change of resistivity with frequency, the so-called frequency-effect (FE), which is a measure of polarization.


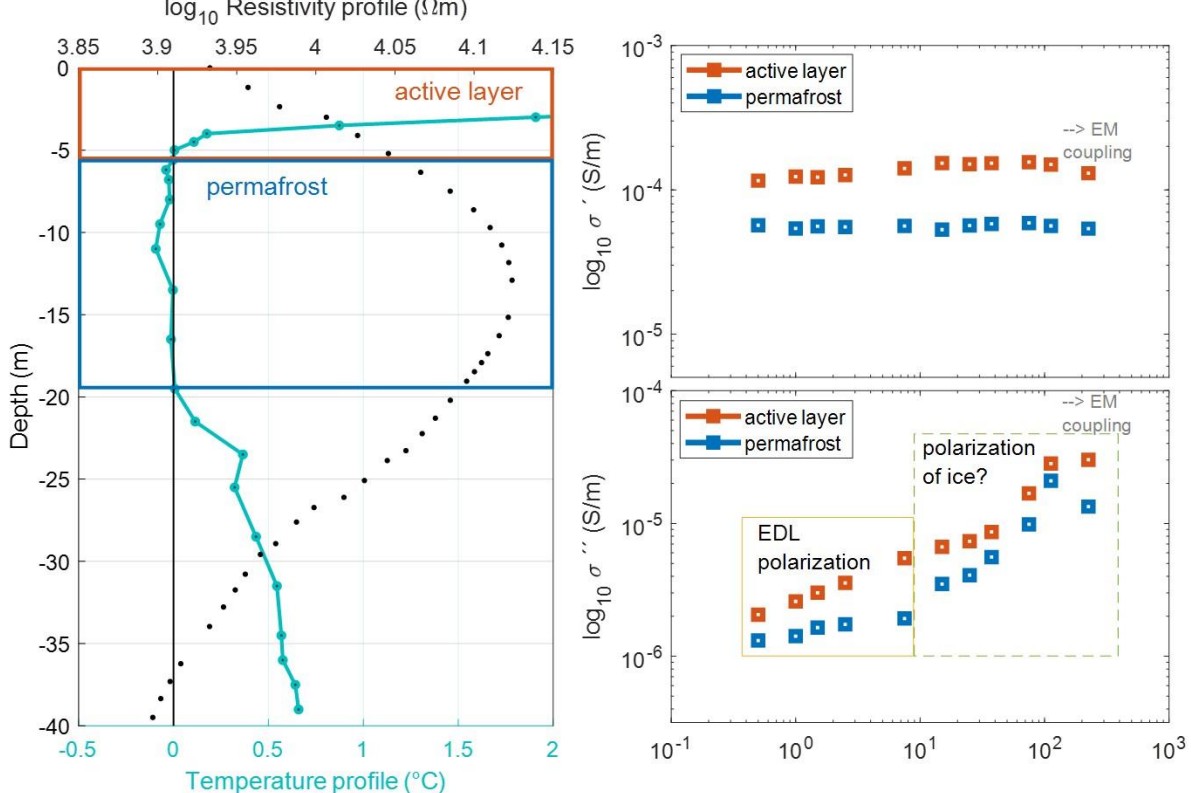

Figure 9: Borehole temperature at BH-1108 at the time of the SIP survey with resistivity profile at close proximity to the borehole (left), and spectral response for a selected depth column at the borehole extracted from the inverted image of profile H2 (right). Pixel values represent the mean value of real and imaginary components and phase of the complex conductivity extracted for two different layers: 0 – 5.6 m depth (active layer) and 5.6 – 19.5 m depth (permafrost).

However, a comparison of these studies with our results is not straight forward, as they investigated the SIP response of different materials ranging from saline permafrost to sandstone and granite samples which are hardly comparable to the SIP response of a talus slope consisting of big blocks and voids filled with ice and air (where mixing models would be needed to upscale the electrical response from the sample scale to the scale resolved by electrical imaging). Additionally, the various permafrost studies use different parameters to describe the frequency-dependence of the polarization, thus care has to be taken in the comparison of the results. Within the frequency range investigated in our analysis, we are not able to capture the ice relaxation peak (ranging from 1 to 100 kHz for relevant permafrost temperatures), hence, in this study we cannot apply the approach of Grimm and Stillman (2015) to quantify ice content. Nevertheless, the increase in polarization magnitude of the ice-rich talus material observed in Fig. 8 is consistent with the observed decrease in temperature and increasing ice content. Given the lack of high-frequency data, this study does not address the distribution of spectral (e.g., Cole- Cole) parameters and their correlation with spatiotemporal changes in the subsurface. However, further studies should consider such analyses together with laboratory measurements in samples to improve the quantification of ice content in alpine permafrost.



# 6 Conclusions and outlook

In this study, we presented SIP imaging data collected in the frequency domain from 0.1 to 225 Hz using a dipole-dipole measurement protocol. We compared data acquired using single multicore, separated multicore, single coaxial and separated coaxial cables to find a field protocol that provides SIP imaging data sets less affected by EM coupling. Results demonstrate that single coaxial cables yield a high reciprocity in the range between 0.5 and 75 Hz leading to an improvement in SNR compared to the other cable setups.

Using this measurement setup we characterized the spatial extent of permafrost at the Lapires talus slope based on the real and imaginary components of the complex resistivity at 0.5 Hz. We observe an overall smaller spatial extent of permafrost compared to the assumed distribution from previous studies and a slight decrease of the spatial extent of permafrost with depth. The complex resistivity images reveal clear variations in the electrical values between frozen and unfrozen states and between different substrates, but more pronounced variations in the pattern of the frequency-dependence of $\rho''$ and $\phi$, compared to $\rho'$. Lowest $\rho', \rho''$ and $\phi$ values were identified for fine-grained unfrozen conditions, while the spectral behaviour was different for ice-rich, blocky material, with increasing $\rho''$ and $\phi$ values with increasing frequency already at 10 Hz. Additionally, our results suggest that SIP data may help to reduce the ambiguity in the interpretation regarding ice-rich, blocky material and unfrozen bedrock, as unfrozen bedrock may be identified based on the local maximum at low frequencies (~10 Hz) in both $\rho''$ and $\phi$ images. This may also help to improve the ice content estimation using petrophysical models.

We conclude that with appropriate measurement and processing procedures, the characteristic dependence of the SIP response of frozen rocks on temperature, and thus ice content, can be utilized in field surveys for an improved assessment of thermal state and ice content at permafrost sites. Future research should be particularly conducted on the combination of field studies with laboratory analyses for an ice content quantification that would rely on SIP data. Further studies should concentrate on the time-lapse analysis of the polarization response in permafrost environments and the application of SIP for different permafrost landforms (e.g. rock glaciers, bedrock permafrost).

*Data availability.* The borehole temperature data from the PERMOS network are available under PERMOS (2019), http://dx.doi.org/10.13093/permos-2019-01. The SIP data that support the findings of this study are available from the corresponding author upon request and will be provided through an online data repository after acceptance of the manuscript.

*Author contributions.* AFO, CHa, CHi and TM designed the experimental setup, TM and AFO collected and processed the geophysical data. AFO, CHa, CHi and TM interpreted the geophysical signatures and AFO, CHa, CHi, AK and TM discussed the results. TM led the preparation of the draft, where AFO, CHa, CHi and AK contributed equally.

*Acknowledgements.* This study is supported by the Swiss National Science Foundation (SNSF) and the German Research Foundation (DFG). We are furthermore thankful to the PERMOS office for providing the borehole temperature data and to the



cable car company Télénendaz S.A. for the logistical support of our research activities at the Lapires field site. We also thank
Martin Mayr, Philipp Zehetgruber and Guy Ramsden for the help in the collection of the geophysical data.

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
