# Peer review of "Spectral Induced Polarization imaging to investigate an ice-rich mountain permafrost site in Switzerland"

_The Cryosphere, 2021_

## Author Comment (AC1)

Authors present an interesting application of SIP on a permafrost site. The study is exhaustive and well presented, results are supported by the data presented and the work is of interest for TC readers. I have no doubt the paper deserves publication after some minor corrections on text and graphics.

Thank you for the constructive and helpful comments. We considered each comment carefully and address them in detail in the following text.

1) Line 73 and 76 seems describe the same concept, maybe the lab tests can be presented in one sentence.

To avoid repeating the description of the laboratory tests, we summarized the concept in one sentence in lines 74-76.

2) line 140  $\rho$  is the here the real component ( $\Omega$  m) ?

We agree that using another word for the real component of the complex resistivity (i.e. the in-phase resistivity) is misleading and removed it from the text.

- **3)** Ln 176 Maybe the relaxation frequency of ice should be here better introduced According to this comment, we specified the relaxation frequency earlier in chapter 2.1 (lines 175-180), so that in line 176 (now lines 184-185), the parameter is already introduced.
- 4) chap. 3.2 The SIP setup description should include essential information as the electrode spacing, the arrays lengths, to etc. to be specified line by line Electrode spacing, profile length and other information about the measurement setup are summarized in Table 1. Nevertheless, we reference this table now more clearly and add the most important details also to the text (Section 3.2.1).
- 5) chap.3.2.2. August or September 2019 ? (see line 218). I think part of this paragraph is setup (3.2) rather than mapping. You assert resistance contact was in the range 5-60 k $\Omega$ , a very low values in such a debris condition (if compared to the literature ones, in the common order of hundreds). Did you help the contact some way ? and if yes, how do you increase the contact locally? This is of extreme importance in SIP results obtained below, and of huge interest for the TC community
  - Thank you for the remark, we wrote the wrong date in line 218 (now lines 223-224) and changed it accordingly.

- Additionally, we changed the titles of subchapters 3.2.1 and 3.2.2 to make clear that all subchapters are part of the SIP measurement setup and describe different aspects of SIP data acquisition.
- The electrode contact was improved by adding salty water containing a small amount of salt in the solution and some mud found in proximity to the profile. This led to a reduction in the contact resistances of a few tens of kΩ, higher current injections and an enhanced IP data quality. We added this description to chapter 3.2 where we specified the improvement of contact resistances.

**6) Ln 258 is it always 5 m the spacing ? did you collect dip dip data also with 10 m spacing?**

We added a sentence clarifying that we collected DD data with 4 m, 5 m and 10 m electrode spacing and different dipole lengths, which we specified in lines 267-269.

**7) In 302-304. This last sentence about robust inversion is not clear, is it real necessary?**

In the presence of data outliers, i.e. errors in DC and complex resistivity measurements due to poor galvanic contact or incomplete removal of electromagnetic coupling, or in the case of unknown error parameters, LaBrecque and Ward (1990) suggest a robust inversion algorithm in which errors of data with large misfits are successively increased during the inversion. Morelli and LaBrecque (1996) tested its efficiency within a DC resistivity tomographic inversion algorithm and Kemna (2000) applied the scheme of LaBrecque and Ward (1990) to complex data errors. Within our analysis, we noticed stable reconstruction results when applying the robust scheme to noisy data (i.e. SIP data collected with multicore cables or in other permafrost environments). Whereas for the use of coaxial cables, where outliers are being minimized we saw no improvement in applying a robust inversion. We tried to better explain this approach in lines 319-325 (revised version).

8) Fig.2, I suggest to label the figures (a,b,c,d) to help reader's comprehension. Modify the text and caption accordingly.

We adapted the figures, caption and text accordingly. Due to the similar structure of Figure 2 and Figure 3, we added labels to both figures.

9) Ln 402. Sentence is not clear, I think the concept is ρa has no frequency dependence but it is still able to discern frozen parts.
 We agree with the reviewer and introduce the term of apparent resistivity in our

manuscript (in lines 146-147) and improved the formulation in lines 424-425 to better transport the message.

**10) Ln 546-549, Sentence about Duvillard work is not clear.**

We clarified the findings of Duvillard et al. (2021) in lines 576-581 as follows: Duvillard et al. (2021) applied a petrophysical model parametrized with a laboratory calibrated freezing curve to conductivity and induced polarization tomography data to assess the temperature distribution of a permafrost-affected rock ridge. They postulate that it only supports ERT data in assessing the contribution of the surface conductivity to the total conductivity of the material. However, they did not investigate the frequency dependence of the IP, and thus, did not exploit polarization processes occurring in another frequency range, as TDIP field data were acquired at a dominant frequency of ~1 Hz.

11) Fig.8 the c) panels must be differentiated, cause they are not clear in the caption. Please label the different depths in the figure.

We modified Fig. 8 and its caption accordingly by adding the labels of the different depths to the legend of Fig. 8a and Fig. 8b, added a title to each subplot and defined a common legend for the whole figure.

12) Ln 584-87 As I understood you performed lab test on Lapires sample rocks. This is part of the research (and then Method) and should no be presented here in the discussion section.

We are investigating here the capabilities of the SIP imaging method for the investigation of alpine permafrost sites. The analysis of the micro-scale polarization processes is not the scope of the study. Thus, we prefer to avoid presenting details on the methodology and discussing details of laboratory SIP measurements. This has been discussed in Limbrock et al. (2021).

13) Ln 601 Since the spacing was 5 m, why you average pixel of 1 m width. Maybe keeping the same spacing address more your survey lateral resolution.
We agree with the reviewer and changed the pixel width corresponding to the electrode spacing for all figures and in the description of the figures (line 485, line 597, line 648, Fig. 7, Fig. 8, Fig. 9).

**14) Ln 614 How Coperey work differs from Limbrock one? This is of interest in the discussion of your results.**

Coperey et al. (2019) investigated the effect of temperature changes on the SIP response of porous media and concentrates on the Stern layer polarization mechanism; while Limbrock et al. (2021) consider the polarization effects at a broader frequency range to include the polarization of ice. To address your comment, we added the

following line in the Introduction Section: "Limbrock et al. (2021) characterized the textural and mineralogical controls on the SIP behavior different rock samples collected in the Alps and analyzed the temperature dependence of its polarization response in the laboratory."

**15) Fig. 9 I suggest label the panel. Looking the figure, the frequency dependence of AL and permafrost seems to have the same behaviour. This should be more emphasised in yours discussion.**

We added the labels to Fig. 9 and adapted the caption and description of the figure accordingly. We agree with the reviewer that the complex conductivity values of the active layer and permafrost example are quite similar and both spectra (active layer and permafrost) show an increase in the polarization response with increasing frequency. However, for frequencies above 10 Hz, we clearly can see an increase of the polarization (i.e. the imaginary conductivity and the phase) with a larger slope for the permafrost quantitatively, we computed the slope for the plase of the complex conductivity between 112.5 Hz and 7.5 Hz. The slope of the permafrost is with a value of ~3 mrad/Hz higher compared to the active layer with ~1.6 mrad/Hz slope. This information was added in the Discussion Section in lines 665-673.

Thank you for the very interesting reading

---

## Author Comment (AC2)

**The paper `Spectral induced polarization imaging to investigate an ice-rich mountain permafrost site in Switzerland` (authors: Teresa Maierhofer, Christian Hauck, Christin Hilbich, Andreas Kemna and Adrián Flores-Orozco) is one of the only very few studies investigating the applicability of the IP method in mountain permafrost. This paper brings valuable technical knowledge in discriminating between permafrost and unfrozen substrate within a talus slope in the Swiss Alps. Furthermore, the field protocol presented in this paper is essential to the permafrost community since its application is expected to provide supplementary information on the permafrost characteristics elsewhere. All the methodological parameters were carefully analyzed, and the results were compared with ERT measurements, borehole data and other studies. Overall, SIP discriminated very well against permafrost and unfrozen bedrock.**

**The paper is clearly structured and well-illustrated, contains adequate critical reflections and reaches appropriate conclusions. The methodology is rigorous and consistent, integrating several complementing techniques, whereas data back the interpretations. The results support the findings. There are no factual errors. Therefore, I suggest acceptance of the manuscript after some minor revisions. These are outlined below:**

Thank you very much for these encouraging words, which we appreciate a lot! We considered your suggestions carefully and address them in the following in detail.

1) **In the Introduction, you mention that there are still limitations of 4PM because of the difficult interpretation of the geophysical signature of air, water, ice etc., based solely on 4PM, and thus, an improvement is desirable. Since SIP appears to be a promising complementary technique, it would be interesting to write a phrase within Section 5 (Discussion) to highlight how/if SIP can overcome these limitations and if this technique might be integrated into a more complex model in the future. However, I feel the potential of this new technique for overcoming the existing limitations was not clear enough discussed.**

   Thank you for this comment. We agree of course very much with it, as integrating SIP data into quantitative assessments of the subsurface ice/water etc. contents is one of the ultimate goals of the project, which this paper is part of. This work, though promising, is still in progress and further analyses are necessary, also with respect to laboratory data. We included the following phrase in Section 5 in lines 700-703: "Concerning the improvement of the 4PM, our results suggest that SIP data may help in the differentiation between air-filled and ice-rich, blocky material; and between frozen and unfrozen bedrock, since air has no polarization response and unfrozen bedrock can be identified based on the local maximum at low frequencies in both $\rho''$ and $\phi$ images (see Fig. 8 and 9). However, further works are required to quantitatively include IP into the 4PM."

2) **It seems that sensor spacing is essential for delineating the active layer. Can you add an idea within the Discussion to refer to the importance of SIP sensor spacing for discriminating between ice rich-permafrost and unfrozen substrate and precisely delineating the active layer based on your experience/other studies findings?**

To demonstrate the importance of the selection of the separation between electrodes for the example of the Lapires talus slope, we show in the revisions (appendix) a numerical modelling example. Based on the IP imaging results at 1 Hz along profile H1 and the nearby borehole information, we generated a synthetic model consisting of a 5m deep active layer ($\rho$ =2.5 kΩm, $\varphi = -5$ mrad) on top of the permafrost body ($\rho$ =25 kΩm, $\varphi = -40$ mrad) for the same electrode configuration chosen as presented in our field study (dipole dipole, skip 3) and for different electrode separations (1m, 2m, 5m, 10m). When comparing the numerical results with the model, we find that a 10m electrode spacing overestimates the active layer thickness, while for the 5m electrode spacing the thickness of the first layer is sufficiently well reproduced for both, resistivity and phase.

We also added the following paragraph to the Experimental Setup section (lines 272-276): "As observed in previous studies (Kemna, 2000; Slater et al., 2000), the spatial resolution of the imaging results depends on electrode separation, measurement schedule, distribution of the electrical subsurface properties and inversion approach (data error). Further details on the selection of an appropriate survey design (i.e. electrode configuration and electrode separation) regarding the resolution of the images can be found in Bing and Greenhalgh (2000), Stummer et al. (2004)."

Additionally, we included the following paragraph in the Discussion section (lines 641-648): "The electrode spacing of 5 m was chosen to improve the resolution close to the surface and our delineation of the contact between the active layer and the permafrost body. Hilbich et al. (2009) applied synthetic modelling and analyzed the depth of investigation (DOI) index (introduced by Oldenburg and Li, 1999) and the resolution matrix (e.g. Menke, 1984) to identify unreliable model regions in ERT data collected at a coarse blocky and ice-rich permafrost site. They found that the determination of the active-layer thickness – transition to the permafrost table with electrical methods is limited by its vertical resolution which is dependent on the electrode spacing. Thus, from numerical modelling (not shown) we found that a separation of 5m with dipole lengths of 5 and 20 meters was small enough to delineate the transition between the active layer and the ice-rich permafrost."

3) **Figure 1 needs some adding. First, there is no scale on the inset map with Switzerland. Then, you should add to the legend the ERT profiles' location. PERMOS monitoring profile also refers to ERT? Please clarify! Moreover, there are no values for the contour lines. Finally, you should mention the source of the background image.**

We agree with all suggestions and changed them accordingly (scale on inset map, PERMOS ERT monitoring profile, values of contour lines, source of background image) except the labeling of the ERT profiles in Fig. 1, because we think that it is not necessary to mention this explicitly as we did not collect additional ERT data, we only applied a DC resistivity inversion (Fig. 4) to the SIP data (SIP profiles H1, H3, H4, V2, V3) shown in Fig. 1. Instead, we mentioned this detail in the caption of Fig. 4.

**4) Please add the coordinates of the Lapires talus slope in Section 3.1.**

We added the coordinates (46°6'22.46" N, 7°17'3.40" E) to Section 3.1.

**5) In section 3.1, you mention that lithology consists of gneiss and schists, but in Section 5.2 (lines 580-585), you say that granite was collected and examined in the Lapires site. It is not consistent; please clarify.**

Thank you for spotting this error. The local geology is dominated by gneiss and schist (as outlined in section 3.1 and indicated in the geological map: Bundesamt für Landestopographie Swisstopo. Geologischer Atlas der Schweiz 1:25000.). The sample that we collected close to the PERMOS monitoring profile was also a gneiss sample. We have edited our manuscript to correct this apparent contradiction in section 5.2, now we say in L617 that the laboratory measurements were conducted in a representative rock sample collected from the site, with a composition of 39% muscovite, 36% quartz, 17% albite, 4% microcline and 4% clinochlore.

**6) Some studies were conducted in the Lapires site, which were not cited and might be needed to consider (e.g., Delaloye et al., 2001; Scapozza et al., 2010).**

We agree with the reviewer and considered the studies in Section 3.1. We added the following missing detail in lines 198 and 207: "During drilling of all boreholes, the bedrock was not reached, as reported by Scapozza et al. (2010)."

**7) Figure 4 should also contain information on the RMS error and electrode configuration. You added the intersection between crossing profiles but didn`t specify the name of the intersected profiles.**

Thank you for the indication. All inversions were fitted to an error-weighted rms value of 1 for the smoothest possible model with the error parameters shown in section 3.3. We added the following description to Section 3.3: "The code allows terminating the inversion when a root-mean-square of 1 is reached, which means fitting the model to the data considering the interval of confidence quantified by means of the error parameters." The electrode configuration for each profile is specified in Table 1, but we adapted the caption of Fig. 4 accordingly, to make it clearer. Additionally, we added a label to the intersection between crossing profiles in Fig. 4.

**8) Geophysical measurements were conducted in two different years. Have you noticed significant changes in the permafrost spatial extent between 2018 and 2019?**

In 2018, the focus of our study was to find a field protocol that provided good-quality SIP data sets. Thus, in 2018, we only tested different cable setups along a small profile, measurements for mapping across 6 profiles were collected in 2019. Hence, the data are discussed separately as they provide different information. While the data in 2018 help to evaluate data quality, the measurements in 2019 are used to identify the spatial distribution of the permafrost. We do not compare the results obtained for measurements in 2018 and 2019 as this is not the scope of the study. The only profile that we collected in both years was the PERMOS monitoring profile, were we had to use the fix-installed multicore cables. The IP data quality for these cables (as can be seen from Fig. 2) was poor, why we cannot give a definite answer to this question. From ERT alone, there are no significant changes in permafrost extent between these two years (PERMOS, 2021). However, regarding the Permafrost monitoring at this site, there is indeed a long-term trend towards decreasing resistivities of the ice-rich permafrost body and increasing permafrost temperatures as observed through the ERT monitoring profile and the boreholes existing at this site (https://newshinypermos.geo.uzh.ch/app/DataBrowser/). The short time span between the measurements in 2018 and 2019 is however not sufficient to resolve this trend, which is also not the scope of this paper.

**9)** **Did the ERT monitoring multicore cable installed within PERMOS ERT network influence your SIP measurements?**

We installed the coaxial cables at a distance of 5-10 meters away from the PERMOS profile. However, the multicore cables would not influence our measurements, since the cables are isolated, thus their influence in the current pathways is negligible.

**10)** **There is a significant reduction in permafrost extent compared with Delaloye (2004). Therefore, in Section 5.2 (lines 530-535), I suggest providing the difference in spatial extent between your findings and Delaloye (2004). On the other hand `the slight decrease of the spatial extent of permafrost with depth` is also supported by findings of Mollaret et al. 2019, which showed that the most significant resistivity decrease at LAH (profile H1) is `below the ice-rich body` due to air circulation. Please refer to this within Section 5.2.**

Thank you for the suggestion, but as already discussed in the lines below (until L576), a direct comparison of the extent by Delaloye 2004 and our findings may be misleading as they are based on different methods and had different objectives. We therefore prefer to avoid a quantitative difference.

However, we think that within our study we cannot conclude on the temporal change of the depth extent, as we only compared the largest extent of permafrost derived from our study with the original polygon. The study of Mollaret et al. (2019) does not investigate the same spatial area in their analysis. Thus, we tried to make it clear, that the observation of the reduction in size with depth was only meant for the measurements conducted in 2019 (lines 561-562). Yet, we added the findings of the study of Mollaret et al. (2019) in the discussion of our results (lines 569-570).

11) **Technical corrections: The reference list lacks information about several papers cited in the text: Limbrock et al., 2021 (line 584), Duvillard et al. 2020 (line 546), Haeberli et al., 2018 (line 33), Krainer et al., 2015 (line 33); Revil et al., 2012 (line 63) (at the reference list is Revil et al., 2012a and 2012b), Waxman et al., 1968, Biskaborn 2016 (line 28) (at the reference list is 2019)? However, I think Biskaborn et al., 2019 is correct. Coperey et al. 2019a (line 43) should be replaced with Coperey et al. 2019 and Scapozza et al., 2015a should be replaced with Scapozza et al., 2015. Flores Orozco et al., 2018 and 2019 are in the Reference list but not cited in the text. The same goes for Haeberli et al., 1988, Krainer et al., 2014, Shengting et al., 2018!, Waxman and Smits, 1968!**

We thank the reviewer for helping to spot this error. We have corrected and reviewed our reference list. We hope we have now listed all references mentioned and vice versa.

References:

Delaloye, R., Reynard, E. & Lambiel, C. (2001). Pergélisol et construction de remontées mécaniques : l'exemple des Lapires (Mont-Gelé, Valais). Le gel en géotechnique, Publications de la Société Suisse de Mécanique des Sols et des Roches, 141, 103- 113.

Scapozza, C., Lambiel, C., Abbet, D., Delaloye, R., Hilbich, C., 2010. Internal structure and permafrost characteristics of the Lapires talus slope (Nendaz, Valais). 8th Swiss Geoscience Meeting 2010, Fribourg, Switzerland, 19–20 November 2010. Extended Abstract 7.16, 166–167

Appendix:

[Figure]

*Figure 1 : Numeric modelling example for different electrode spacings (1m, 2m, 5m and 10m). The synthetic model is based on the IP imaging results at 1 Hz along profile H1 collected with a dipole dipole configuration and the nearby borehole information.*